# MALDI-MS Analysis of Peptide Libraries Expands the Scope of Substrates for Farnesyltransferase

**DOI:** 10.3390/ijms222112042

**Published:** 2021-11-07

**Authors:** Garrett L. Schey, Peter H. Buttery, Emily R. Hildebrandt, Sadie X. Novak, Walter K. Schmidt, James L. Hougland, Mark D. Distefano

**Affiliations:** 1Department of Medicinal Chemistry, University of Minnesota, Minneapolis, MN 55455, USA; schey013@umn.edu; 2Department of Chemistry, University of Minnesota, Minneapolis, MN 55455, USA; peterhb@live.unc.edu; 3Department of Biochemistry and Molecular Biology, University of Georgia, Athens, GA 30602, USA; erh@uga.edu (E.R.H.); wschmidt@uga.edu (W.K.S.); 4Department of Chemistry, Syracuse University, Syracuse, NY 13244, USA; sxnovak@syr.edu (S.X.N.); hougland@syr.edu (J.L.H.); 5BioInspired Syracuse, Syracuse University, Syracuse, NY 13244, USA

**Keywords:** protein prenylation, enzymology, MALDI libraries

## Abstract

Protein farnesylation is a post-translational modification where a 15-carbon farnesyl isoprenoid is appended to the C-terminal end of a protein by farnesyltransferase (FTase). This modification typically causes proteins to associate with the membrane and allows them to participate in signaling pathways. In the canonical understanding of FTase, the isoprenoids are attached to the cysteine residue of a four-amino-acid CaaX box sequence. However, recent work has shown that five-amino-acid sequences can be recognized, including the pentapeptide CMIIM. This paper describes a new systematic approach to discover novel peptide substrates for FTase by combining the combinatorial power of solid-phase peptide synthesis (SPPS) with the ease of matrix-assisted laser desorption ionization-mass spectrometry (MALDI-MS). The workflow consists of synthesizing focused libraries containing 10–20 sequences obtained by randomizing a synthetic peptide at a single position. Incubation of the library with FTase and farnesyl pyrophosphate (FPP) followed by mass spectrometric analysis allows the enzymatic products to be clearly resolved from starting peptides due to the increase in mass that occurs upon farnesylation. Using this method, 30 hits were obtained from a series of libraries containing a total of 80 members. Eight of the above peptides were selected for further evaluation, reflecting a mixture that represented a sampling of diverse substrate space. Six of these sequences were found to be bona fide substrates for FTase, with several meeting or surpassing the in vitro efficiency of the benchmark sequence CMIIM. Experiments in yeast demonstrated that proteins bearing these sequences can be efficiently farnesylated within live cells. Additionally, a bioinformatics search showed that a variety of pentapeptide CaaaX sequences can be found in the mammalian genome, and several of these sequences display excellent farnesylation in vitro and in yeast cells, suggesting that the number of farnesylated proteins within mammalian cells may be larger than previously thought.

## 1. Introduction

Protein prenylation is a post-translational modification which involves the covalent attachment of a hydrophobic isoprenoid group to the thiol side chain of a cysteine residue located near the C-terminus of a protein. Farnesyltransferase (FTase) and geranylgeranyl transferase (GGTase-I) transfer a 15-carbon farnesyl and a 20-carbon geranylgeranyl group, respectively [1]. These enzymes recognize proteins with a C-terminal tetrapeptide consensus sequence known as a “CaaX box”, where “C” is the cysteine residue that is covalently modified, “a” is usually an aliphatic amino acid, and the “X” is a residue that is largely responsible for determining whether the protein substrate is targeted by FTase or GGTase-I [2] (Figure 1). Protein prenylation is essential for proper cellular localization and signaling activity, and misregulation of prenylated proteins is implicated in many diseases [3,4]. For this reason, prenylation has also drawn more recent attention as a potential target for the treatment of Alzheimer’s disease, Hutchinson−Gilford progeria syndrome, and numerous other diseases [5,6,7,8,9]. In 2020, the first inhibitor of FTase, lonafarnib, was approved for the treatment of progeria [10,11].

FTase manifests broad substrate specificity, catalyzing the transfer of a farnesyl group from farnesyl pyrophosphate (FPP) to a variety of polypeptide substrates, and many attempts have been made to define what amino acids are preferred or not preferred in the CaaX sequence [12,13]. This promiscuity has even been leveraged in the design of novel mutant FTases for orthogonal labeling [14]. While the canonical model of the CaaX box is generally well understood, it has recently been found that certain sequences longer than the four-residue CaaX motif can also be farnesylated by both yeast and mammalian FTase orthologs [15]. These CaaaX motifs were first observed in yeast, and initial evaluation of CaaaX substrate space found the sequence CMIIM to be the prototype for the CaaaX sequence, with in vitro assays indicating that this peptide was a reasonable substrate (k_cat_/K_M_ = 1.9 × 10^4^ M^−1^s^−1^), although less efficient compared with the most efficient CaaX peptides such as CVLS (k_cat_/K_M_ = 2.0 × 10^5^ M^−1^s^−1^) [15].

Though the substrate space of the CaaX-containing peptides has become increasingly well defined as a result of the past three decades of research, there are many questions remaining about how this information might apply to the extended CaaaX sequences. Work in our lab and others has relied on a library-based screening approach to probe the peptide substrate space, which involves generating a large number of peptides that have a systematic combination of amino acids. Previous analysis of peptide libraries has involved the use of an alkyne-containing isoprenoid analogue to allow for biotin attachment by derivatizing with biotin-azide via copper-catalyzed azide−alkyne cycloaddition [16]. The attached biotin then allowed for the visualization of farnesylated peptides via an enzyme-linked assay involving streptavidin-alkaline phosphatase to form a colored product. One disadvantage of that approach is that it relies on the use of synthetic isoprenoid analogues that may perturb enzyme specificity. To complement that approach, we developed an alternative peptide library screening strategy to analyze the substrate space of pentapeptide CaaaX sequences that would employ MALDI-MS as the method of detection. Since peptide farnesylation results in a significant increase in mass, farnesylated products are easily separated from their unfarnesylated precursors. Moreover, MALDI typically generates singly charged species without fragmentation, allowing for rapid sample analysis and high sensitivity, and is amenable to complex mixtures, giving it numerous advantages [17,18]. Thus, it was hypothesized that MALDI would allow for libraries containing 10–20 members to be quickly and easily analyzed while utilizing the native substrate FPP without the need for a biorthogonal analog for subsequent visualization. Splitting libraries into 2 sets with 10 of the canonical amino acids allows for analysis without isotopic overlap of Leu/Ile and Lys/Gln [19,20]. To start this exploration of the novel substrate space, we utilized peptide libraries based on randomization of the best characterized CaaaX sequence, CMIIM, to determine if this sequence’s farnesylatability could be improved, and if additional amino acid substitutions not normally considered canonically “CaaX-like” could be identified.

## 2. Results

### 2.1. Validation and Optimization of MALDI Method with Known Substrates

To determine whether the proposed MALDI method could be adapted to identify novel CaaaX sequences, initial efforts focused on peptides with the sequence DsGRAGCVa_2_A (where Ds is a dansyl group). The canonical CaaX tetrapeptide CVIA is a native substrate for the yeast FTase, and libraries examining the sequence variability at the a_2_ and X positions have been previously reported, making this an excellent test case. A DsGRAG tag was appended onto the N-terminus to aid in purification and increase ionization efficiency [21,22]. Thus, a 17-membered DsGRAGCVa_2_A library was synthesized, where X was varied to all 20 proteogenic amino acids except cysteine, leucine, and glutamine. Cysteine was omitted due to potential synthetic difficulties, while leucine and glutamine were omitted as they have nearly identical monoisotopic residue masses to isoleucine and lysine, respectively, and thus would be indistinguishable. In the unreacted library, all individual peptide peaks could be observed and resolved in the MALDI spectrum, including those with a difference of only one mass unit (Figure 2A). This library was farnesylated in the presence of different concentrations of FTase to determine the optimal conditions for farnesylating a complex mixture of substrates. Initial reactions containing 0.1 nM enzyme showed no appreciable product formation (Figure 2B). When the enzyme concentration was increased to 10 nM, the intensity of the unfarnesylated peptides decreased in the MALDI spectrum, and several farnesylated peptides were observed with easily detectable intensity (Figure 2C).

Gratifyingly, most of these initial product peptides contained amino acids at the X position, which were shown to be farnesylated effectively in previous studies [12,16,23]. Increasing the enzyme concentration to 100 nM yielded a similar reduction in the intensity of all unfarnesylated peptides, and 10 farnesylated peptides were observed with remarkably high intensity (Figure 2D).

### 2.2. Identification of Novel Substrates from the CMIIM Motif Using MALDI Analysis

With the above validation complete for a simple CaaX library, several libraries were prepared based on the previously reported pentapeptide CaaaX box CMIIM, where the four positions following cysteine were individually varied to all 20 proteogenic amino acids. This was done using two libraries of 10 peptides for each position, so that all possible amino acid substitutions could be evaluated without the overlap of amino acids with near-identical molecular masses. Thus, eight libraries were created in total, two each for the sequences Ca_1_IIM, CMa_2_IM, CMIa_3_M, and CMIIX. The predicted masses of the starting peptides and products along with their experimentally observed counterparts are provided in Appendix A. While the ionization of the first unfarnesylated CXIIM library was sufficient, initial attempts to farnesylate this library with 100 nM yFTase proved unsuccessful. This was not surprising since CMIIM was reported to be a poorer substrate relative to the native CVIA sequence by a factor of 10. Increasing the enzyme concentration to 1 µM gave several hits (Figure 3B,D), while a reaction containing 10 µM enzyme did not greatly enhance the number of product peptides or their intensity (Appendix A). Therefore, all farnesylation reactions studying the CaaaX libraries were performed using 1 µM yFTase. One potential cause of the need for increased enzyme for these pentapeptide libraries is that in addition to being poorer substrates, some may actually act as inhibitors of FTase. Peptides and peptidomimetic inhibitors of FTase have been known for many years [24]. Gratifyingly, the farnesylation of the CXIIM library showed farnesylation of the parent peptide, as well as several new amino acids. In total, 30 new potentially farnesylated CaaaX sequences were observed using a threshold signal-to-noise ratio of >12 across the eight different libraries (Table 1). Most of the hits in the pentapeptide CMIIX library—including residues Q, A, M, and S—were previously found to be good substrates when in the X position of the corresponding tetrapeptide CaaX box. Approximately half of known X-position amino acids known to be substrates (in the context of the tetrapeptide CaaX box) were observed in our analysis of the CMIIX library. Overall, the X-position libraries contained the lowest number of unexpected (or non-canonical) amino acids, leading us to hypothesize that recognition of the C-terminal residue in a pentapeptide in the enzyme active site may involve similar interactions as occurs with tetrapeptide CaaX box sequences. In contrast, some more unexpected amino acids were detected, including Tyr or His in the a_1_ position. Eight of the above peptides were selected for further evaluation, reflecting a mixture that represents a sampling of diverse substrate space.

### 2.3. Evaluation of Individual Peptide Hits by HPLC

To confirm the farnesylation of the hits in the library-based MALDI analysis and to compare the farnesylation efficiencies of the various peptides with CMIIM, it was necessary to develop a secondary assay. This was readily accomplished by capitalizing on the presence of the dansyl fluorophore present in all of the peptide sequences, using high-pressure liquid chromatography (HPLC) to separate the starting peptides from their cognate farnesylated products. The extent of conversion was quantified by observing the loss of starting material over time via integration of the fluorescence signal. Since these reactions were performed with pure individual peptides, the possibility of inhibition by other nonsubstrate peptides was eliminated. Hence, the concentration of enzyme was reduced from 1 µM to 25 nM. In each case, the structure of the farnesylated peptide was confirmed by liquid chromatography MS2 (LC-MS/MS) analysis. Of particular note, characteristic b4 and b5 ions were used to establish S-farnesylation on cysteine (Appendix A and Appendix A). The percent farnesylation of these peptides is summarized in Table 2. The CMIIM peptide displayed 56% conversion to product in 30 min with 25 nM yeast FTase (yFTase) (Figure 4A). Peptides considered to be similar to the related “CaaX-like” sequences, including CMIIQ, CMIIS, CSIIM, and CMKIM, showed conversion similar to or greater than that of the parent CMIIM (Figure 4B, Appendix A). The sequences CMIGM and CMIIK both displayed little to no conversion under these reaction conditions; this was not particularly surprising since both Gly and Lys are not normally observed in related CaaX sequences in the aliphatic or X position, respectively (Appendix A). We speculate that the presence of the additional cationic lysine residue may boost the ionization of that farnesylated peptide in the initial MALDI screening, even if present in very low abundance. Interestingly, some conversion was observed for the sequence CHIIM, and high conversion was observed for CYIIM, which contain residues that are also not commonly found in farnesylated CaaX sequences (Appendix A). These initial findings indicate that while many of the canonical rules that define CaaX sequences also apply to the extended CaaaX sequence, the extra amino acid allows for some additional flexibility in allowable amino acids and expands the scope of these FTase substrates.

Overall, of the eight peptide sequences that were selected for analysis in the secondary HPLC assay, six (75%) were confirmed as substrates. As a first step to querying the potential relevance of CaaaX sequences to a mammalian system, the above peptides from the yFTase screen were evaluated with rat FTase (rFTase) (Table 2). The only peptides that showed significant conversion with 200 nM rFTase were the CMIIS and CMIIQ sequences. This is not particularly surprising, as the rat enzyme is generally more stringent in what substrates it accepts compared with the yeast enzyme [16,23].

### 2.4. CaaaX Hits in the Mammalian Genome

When searching the human genome for CaaaX sequences of the type CSXXX, CXXXQ, or CXXXS, 138 C-terminal CaaaX sequences were found that exist on known proteins. In order to narrow down this list, those which had unlikely motifs such as multiple charged residues or Gly and Pro amino acids were omitted. That left 28 sequences that might serve as prenyltransferase substrates (Appendix A). Of those sequences, 10 were selected for further evaluation. The results of HPLC assays with those 10 are shown in Table 3. While many of these sequences showed very limited activity with 100 nM rFTase, CMTSQ (Figure 4C) and CASQS (Figure 4D) showed substantial conversion, with CSLMQ showing excellent conversion (Figure 5A). CSLMQ still showed high conversion with as little as 25 nM rFTase, and when this peptide was compared to the native CaaX sequence CVLS at the same enzyme concentration, the results were almost identical, with both peptides achieving 85% conversion (compare Figure 5A,B). Thus, it is striking that an extended CaaaX sequence can be farnesylated as efficiently as a native CaaX sequence, since even the best previously described pentapeptide CaaaX sequence, CMIIM, is approximately an order of magnitude worse compared with the native tetrapeptide ones [15]. These mammalian sequences were also evaluated with yFTase, and the three aforementioned peptides manifested excellent conversion with that enzyme as well, with CSQAS and CSLMQ still being efficiently farnesylated with as little as 25 nM yFTase, with CMTSQ displaying 60% conversion with yFTase, while it had substantially less activity with the rat enzyme. CASSQ also displayed good conversion with the yeast enzyme, which is perhaps unsurprising due to its similarity to the native CaaX sequence CASQ.

### 2.5. Farnesylation of CaaaX Sequences Can Occur Efficiently in Cells

While many of the above peptides were shown to be farnesylated in vitro, an important question concerns their ability to be farnesylated under cellular conditions. Accordingly, to determine whether these pentapeptide CaaaX sequences could be farnesylated in vivo, they were analyzed in the context of the yeast HSP40 protein Ydj1p, which has proven to be a useful reporter system for studying the specificity of farnesyltransferase in yeast [25,26]. Farnesylation of Ydj1p alters its mobility in SDS-PAGE such that farnesylated wild-type Ydj1p (CASQ) has increased mobility (i.e., smaller apparent kDa) relative to unfarnesylated Ydj1p-SASQ [25]. This mobility shift is entirely attributable to farnesylation because the shift is eliminated in the absence of FTase activity as determined using a yeast knockout strain (Appendix A).

A similar FTase-dependent mobility shift was previously used to demonstrate the farnesylation of the reporter protein Ydj1p-CMIIM, which bears a C-terminal pentapeptide CaaaX sequence [15]. By comparison to Ydj1p-SMIIM (a non-farnesylated protein), all of Ydj1p-CMIIM appeared to be shifted to increased mobility, indicating that this and possibly other non-canonical-length CaaaX sequences are able to undergo near-complete farnesylation in cells. Individual constructs containing the CaaaX sequences obtained from the initial library screening described here, fused to Ydj1, were similarly transformed into yeast and evaluated using this mobility shift assay and were determined to be farnesylated to varying degrees (Figure 6). Quantification of the farnesylated and unfarnesylated species in each lane (Appendix A) indicated that Ydj1p-CMIIQ, -CMKIM, -CSIIM, and -CYIIM appeared to be extensively farnesylated (100%, 99%, 100%, and 100%, respectively), -CHIIM and -CMIIS were mostly farnesylated (77% and 89%, respectively), and -CMIGM was modestly farnesylated (40%). In addition, several of the CaaaX sequences observed in the mammalian genome were effectively farnesylated with -CSLMQ showing extensive farnesylation (97%) and -CLLFS and -CSKLN showing more limited farnesylation (16% and 13%, respectively) (Figure 7). A complete summary of this data along with information on biological and technical replicates is provided in Appendix A. Based on these in vivo results, it appears that the scope of farnesylatable sequences is larger than previously believed.

## 3. Discussion

This work describes a new systematic approach to discover novel peptide substrates for FTase by combining the combinatorial power of solid-phase peptide synthesis (SPPS) with the ease of MALDI-MS. The workflow consists of synthesizing focused libraries containing 10–20 sequences obtained by randomizing a synthetic peptide at a single position. As SPPS relies on iterative coupling reactions to extend a peptide from a solid support, the preparation of libraries is quite facile. Incubation of the library with FTase and FPP followed by MALDI analysis allows the enzymatic products to be clearly resolved from starting peptides due to the increase in mass that occurs upon farnesylation (a 204.1 Dalton increase). Positive hits are then confirmed using a secondary HPLC-based assay with purified individual peptides functionalized with a dansyl fluorophore to facilitate quantification. Finally, expression of the yeast protein Ydj1p containing specific C-terminal CaaaX sequences in yeast followed by Western blot analysis of cellular lysates allows the extent of farnesylation within live cells to be ascertained.

In analyzing the results from the extended CaaaX libraries, we found that the a_1_ position manifested the greatest number of hits (10), likely because it is furthest away from the C-terminal X residue that plays a key role in substrate recognition. Numerous hits were also observed in the a_2_ and a_3_ positions, with the a_3_ position showing more canonical (canonical in the context of the tetrapeptide CaaX box) hydrophobic residues (Ala, Leu, and Met). The X position, considered the most important for peptide recognition, showed canonical hits (Ala, Cys, Ser, Gln, and Met) with one exception, Lys, which was not a bona fide substrate of the enzyme when evaluated in HPLC assays.

The results presented here illustrate that many pentapeptide sequences are substrates for FTase. The list of those sequences includes motifs present on the C-termini of bona fide mammalian proteins including transcription elongation factor A protein 3 (CSLMQ), Xaa-Pro aminopeptidase 3 (CSQAS), and beta-1,4 N-acetylgalactosaminyltransferase 1 (CMTSQ). While the data reported herein do not establish that those proteins are in fact farnesylated in vivo, the results imply that they could be. This in turn suggests that the number of farnesylated proteins within cells may be larger than previously thought. As a final point, it should be noted that the workflow presented here could be applied to other post-translational modifications, including lipidation, glycosylation, and others where a significant mass increase occurs. While it might be possible to achieve similar results using ESI-MS in lieu of MALDI-MS, the presence of multiply charged species and various salt adducts complicates the analysis of library-based mixtures when employing the former technique. Although MALDI-based screening has its limitations as a quantitative tool, it is particularly well suited to the type of experiments reported here, where post-translational modification is accompanied by a significant mass shift.

## 4. Materials and Methods

### 4.1. Library Synthesis

Peptide libraries were synthesized using Fmoc-based solid-phase peptide synthesis (SPPS) on a Gyros Protein Technologies AB (Uppsala, Sweden) PS3^®^ peptide synthesizer using four equivalents of Fmoc-protected amino acids from Aldrich^®^ (St. Lous, MO, USA), Novabiochem^®^(Burlington, MA, USA), and P3 Biosystems^®^(Louisville, KY, USA) and Fmoc-AA-Wang resins from P3 Biosystems. A one-pot synthesis of 10 amino acids per library was performed at the “X” position, varying the ratio of amino acids to account for coupling efficiencies [27]. Dansylglycine (DsG) was then coupled manually using a twofold molar excess, reacting for 4–6 h. Following synthesis, peptides were cleaved from resin for 2 h with 5 mL reagent K (82.5% trifluoroacetic acid (TFA), 5% thioanisole, 5% phenol, 2.5% 1,2-ethanedithiol, and 5% H_2_O, v/v) cleavage cocktail per 0.1 mmol of resin. Peptides were precipitated by draining the cleavage cocktail into 40 mL Et_2_O cooled in an isopropanol/dry ice bath for ~10 min, centrifuging until pelleted, decanting the Et_2_O layer, and repeating once to wash the residual cleavage cocktail from the crude peptide mixture. The peptides were then dissolved in 50:50 CH_3_CN/H_2_O containing 0.1% TFA. The total peptide concentration was determined by diluting in 1,4-dioxane and measuring the absorptivity at 338 nm using a Cary (Santa Clara, CA, USA) 50 Bio UV-Visible spectrophotometer, using Beer–Lambert’s law in conjunction with dansylglycine’s molar extinction coefficient (4300 cm^−1^ M^−1^). Individual peptide hits were synthesized using similar conditions.

### 4.2. Enzymatic Farnesylation of Peptide Libraries

Enzymatic farnesylation of peptide libraries was performed by incubating FTase from *S. cerevisiae* (yFTase) in a reaction buffer that contained 20 μM total peptide, 40 μM FPP, 50 mM Tris-HCl pH 7.5, 10 μM ZnCl_2_, 5 mM MgCl_2_, and 1 mM DTT in H_2_O. Reactions were allowed to proceed for 5 h at 37 °C. Upon completion, the samples were desalted using a Plus long Sep-Pak reverse-phase C18 environmental cartridge from Waters Corporation (WAT023635, length 3 cm, diameter 1 cm)(Milford, MA, USA). Sep-Paks were primed by washing with 3 mL Buffer B (CH_3_CN with 0.1% TFA) and equilibrated with 3 mL Buffer A (H_2_O with 0.1% TFA). The sample was then loaded, washed with 2 mL each of 100% Buffer A, 10% Buffer B in Buffer A, and 20% Buffer B in Buffer A, and then eluted with 2 mL Buffer B. Samples were immediately spotted on a MALDI plate or stored at −80 °C. Control libraries were prepared under identical conditions, without the addition of FTase.

### 4.3. MALDI-TOF MS of Farnesylated Peptide libraries

Samples eluted from the Sep-Pak columns (0.5 μL) were cospotted with 0.5 μL of 10 mg/mL α-cyano-4-hydroxycinnamic acid (CHCA) matrix in 50:50 Buffer A–Buffer B on an AB Sciex 384 Opti TOF plate. The typical spotting procedure involved spotting the matrix first, then immediately spotting the sample on top of the matrix, rapidly pipetting up and down to mix. Samples were then analyzed with an AB Sciex 5800 13 MALDI-TOF mass spectrometer (Framingham, MA, USA) using the reflector positive mode. A laser intensity of ~4000–5000 was applied, with a pulse rate of 400 Hz. Laser intensity was increased in increments of 200 if signal was not readily apparent. Four thousand laser shots were applied per spectrum, and the entire spot surface was sampled.

### 4.4. HPLC-Based Enzymatic Farnesylation Assay

Enzymatic farnesylation reactions with purified peptides were performed by incubating FTase from *S. cerevisiae* (yFTase) or *R. norvegicus* (rFTase) in a reaction buffer that contained 2.4 μM peptide, 10 μM FPP, 50 mM Tris-HCl (pH 7.5), 10 μM ZnCl_2_, 5 mM MgCl_2_, and 1 mM DTT in H_2_O [28,29]. Reactions with yFTase were at room temperature (20 °C) for 30 min, reactions with rFTase were run at 35 °C for 45 min. The reactions were flash frozen to stop enzymatic activity, and 200 µL aliquots were injected onto an Agilent 1100 HPLC instrument equipped with an FLD detector (Santa Clara, CA, USA) and a Phenomenex (Torrance, CA, USA) Luna 5 µm C18 100 A pore size 250 × 4.60 mm analytical column. Fluorescence of the dansylated peptides was monitored with an excitation of 220 nm and an emission of 495 nm with a PMT gain of 12. All reactions were run in triplicate. The extent of peptide farnesylation was quantified by integration of the starting material peak from the HPLC chromatogram. The identities of the starting peptides and farnesylated products were confirmed by LC-MS/MS analysis using a ThermoFisher (Waltham, MA, USA) LTQ Orbitrap Velos instrument.

### 4.5. Peptide Search of the Human Proteome

The ScanProsite tool of Expasy was used to scan the UniProtKB for known protein sequences that contain a potential pentapeptide CaaaX sequence (https://prosite.expasy.org/scanprosite/) (accessed on 17 September 2020). The search was limited to C-terminal sequences representative of some of our best peptide hits; the queries searched were CSXXX, CXXXQ, and CXXXS, where any amino acids were allowed in the varied X positions. The scan was performed as a motif search against the UniProtKB, including isoforms. Results were then filtered to only show sequences specific to the human proteome (*H. sapiens*).

### 4.6. Yeast Strains and Plasmids

The yeast strains used in this study have been previously described (Appendix A) [30]. Yeast strains were propagated at rt in either YPD or appropriate selective media when plasmid transformed. Introduction of plasmids into yeast strains was accomplished via a lithium acetate-based transformation procedure [31]. Several of the plasmids used in this study have also been previously described (Appendix A). Others were created in vivo by recombinational cloning using similar methods. In brief, yeast cells were co-transformed by the lithium acetate-based procedure with DNA fragments derived from a NheI and AflII restriction digest of parent plasmid pWS1132 and a DNA fragment encoding the desired CaaaX sequence that was created by PCR using a high-fidelity polymerase. The PCR product encoding the CaaaX sequence was flanked by 5′ and 3′ sequences that were identical to regions of the parent plasmid to facilitate homologous recombination to repair the gapped parent plasmid. Candidate plasmids were recovered from yeast, transformed into and amplified in *E. coli*, and evaluated by restriction digest and commercial DNA sequencing to confirm the presence of the desired YDJ1-CaaaX open reading frame.

### 4.7. Mobility Shift Analysis of Ydj1p Farnesylation

Whole-cell lysates of late-log yeast were prepared as previously described, separated by large-format (19.5 × 16 mm) SDS-PAGE (9.5%), transferred onto nitrocellulose, and blots incubated with rabbit anti-Ydj1p primary antibody (courtesy of Dr. Avrom Caplan) and HRP-conjugated goat anti-rabbit secondary antibody (Kindle Biosciences, Greenwich, CT, USA) [32]. After development of blots with the WesternBright TM ECL-spray (Advansta, San Jose, CA, USA), protein bands were detected using a KwikQuant Imager at multiple exposure times. Levels of farnesylation were quantified from KwikQuant images using ImageJ software for multiple replicates (see Appendix A).

## Figures and Tables

**Figure 1 ijms-22-12042-f001:**
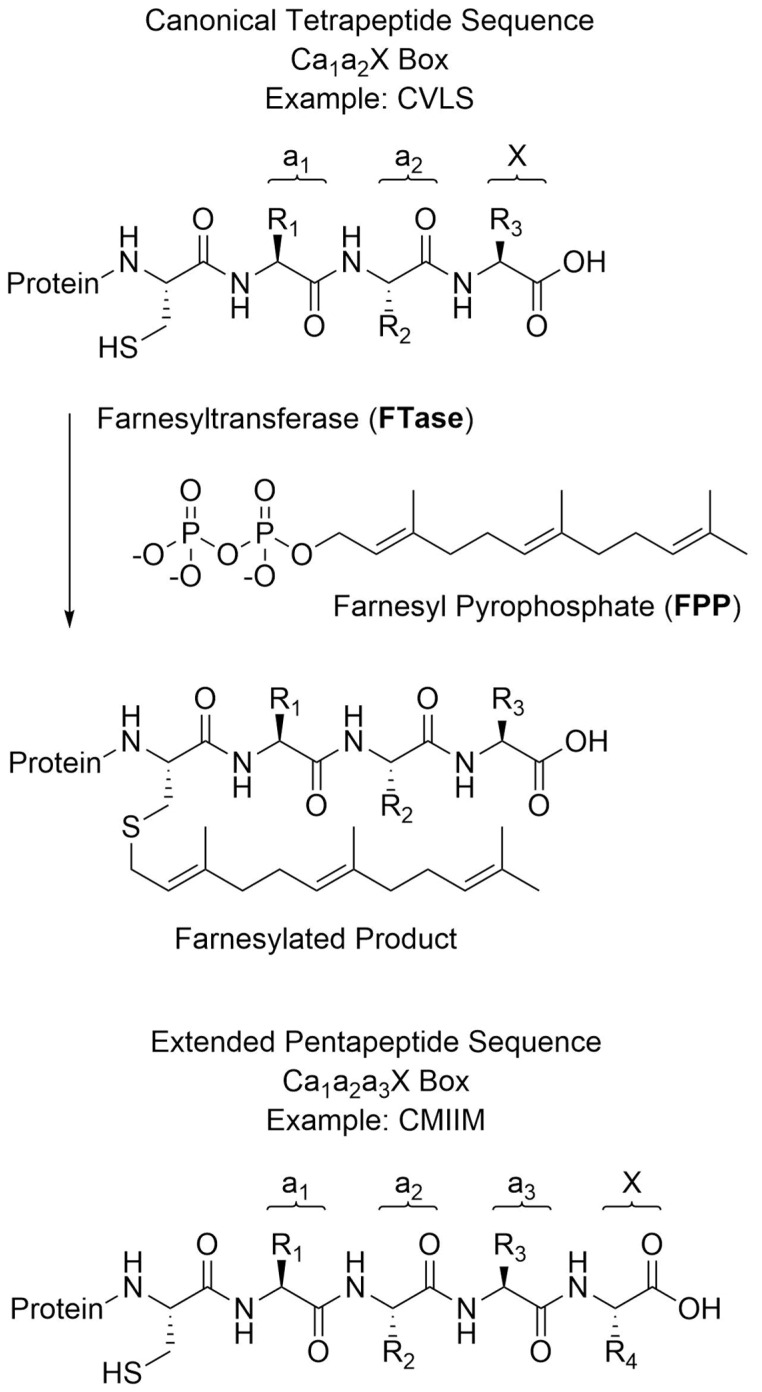
Diagram of the farnesylation of a C-terminal canonical tetrapeptide by FTase, as well as an example of an extended pentapeptide sequence.

**Figure 2 ijms-22-12042-f002:**
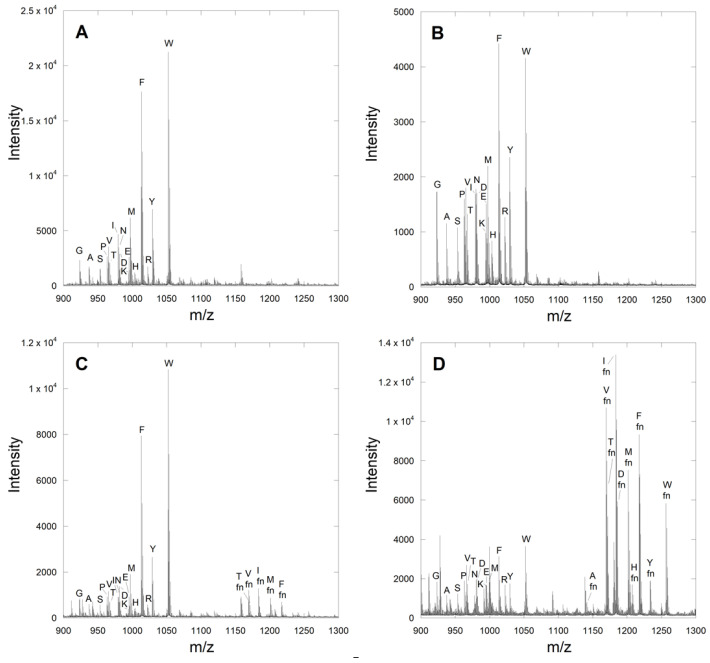
Farnesylation of a DsGRAGCVa_2_A peptide library with varying yFTase concentrations. Libraries reacted with (**A**) no enzyme; (**B**) 0.1 nM enzyme; (**C**) 10 nM enzyme; and (**D**) 100 nM enzyme. The identity of the residue in the X position is indicated with the letter above each peak. The farnesylated peptides are highlighted with the designator “fn”.

**Figure 3 ijms-22-12042-f003:**
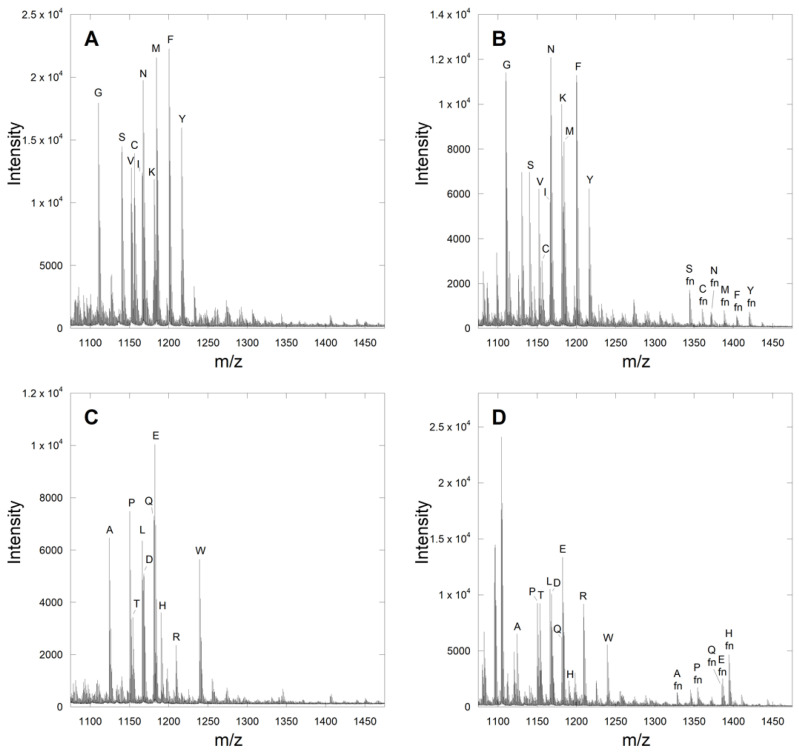
Farnesylation of DsGRAGCa_1_IIM libraries. Library 1 (**A**) before and (**B**) after farnesylation with 1 µM yFTase. Library 2 (**C**) before and (**D**) after farnesylation with 1 µM yFTase. The identity of the residue in the X position is indicated with the letter above each peak. The farnesylated peptides are highlighted with the designator “fn”.

**Figure 4 ijms-22-12042-f004:**
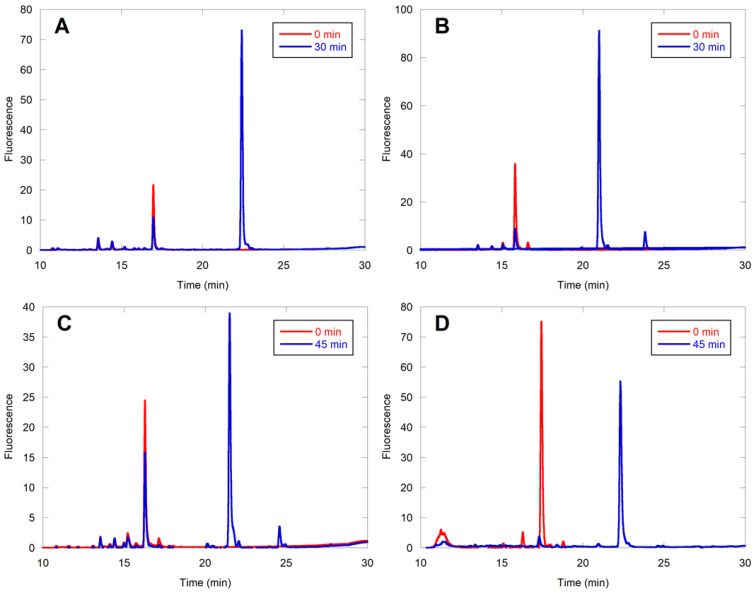
HPLC assays quantifying the conversion of selected peptides by the fluorescence of dansylglycine (ex. 220 nm/em. 495 nm). Reactions contained 2.4 µM peptide, 10 µM FPP, and specified FTase concentration. (**A**) DsGRAGCMIIM (25 nM yFTase); (**B**) DsGRAGCMIIQ (25 nM yFTase); (**C**) DsGCMTSQ (100 nM yFTase); (**D**) DsGCSQAS (100 nM rFTase). Chromatograms of the reaction before (red) and after enzymatic reaction (blue) are shown for each peptide. The farnesylated peptides always eluted later than their unfarnesylated counterparts. It should be noted that CMIIM and CMIIQ are sequences obtained from initial screening of peptide libraries via MALDI whereas CMTSQ and CSQAS are sequences identified from bioinformatic analysis of human genome sequences, guided by the results from the library screening.

**Figure 5 ijms-22-12042-f005:**
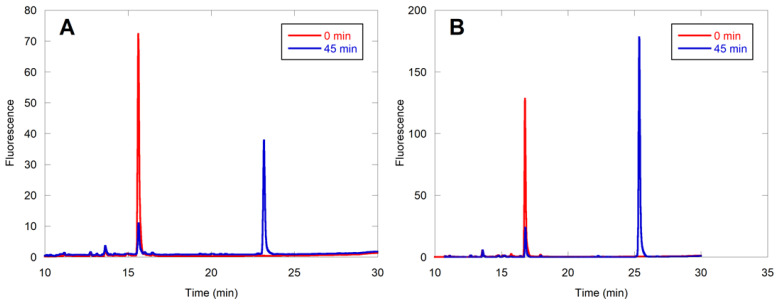
HPLC assays displaying that the best peptide from the bioinformatic screen (CSLMQ) was able to show near-identical conversion to a native tetrapeptide sequence CVLS (85% vs. 84%, respectively). The quantification was performed as in Figure 4. Reactions contained 2.4 µM peptide, 10 µM FPP, and 25 nM enzyme. (**A**) DsGRAGCSLMQ with 25 nM rFTase; (**B**) DsGCVLS with 25 nM yFTase.

**Figure 6 ijms-22-12042-f006:**
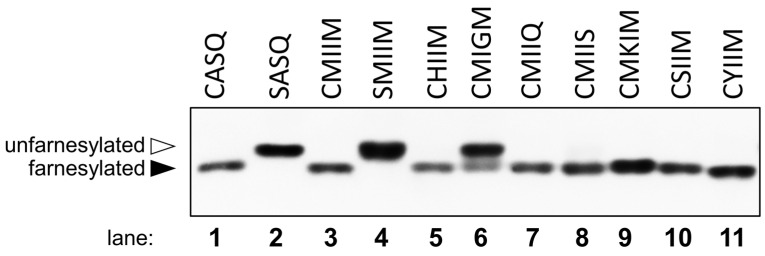
Mobility shift analysis of Ydj1p-CaaaX variants identified from peptide libraries. Whole-cell lysates prepared from yeast expressing the indicated Ydj1p-CaaaX variant were evaluated by SDS-PAGE and anti-Ydj1p immunoblot. The indicated Ydj1p variants were expressed in yWS2544 (ydj1::KANR) to eliminate any contribution from naturally encoded Ydj1p. Farnesylated Ydj1p (CASQ) exhibits a smaller apparent molecular mass relative to unmodified Ydj1p (SASQ). Farnesylation profiles for the indicated Ydj1p-CaaaX variants were determined across multiple biological and technical replicates, from which the percent of farnesylated species relative to the total signal for a sample was determined (see Appendix A and Appendix A).

**Figure 7 ijms-22-12042-f007:**
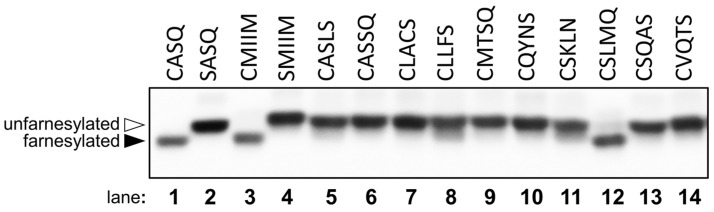
Mobility shift analysis of Ydj1p-CaaaX variants identified from analysis of mammalian genome. Whole cell lysates prepared from yeast expressing the indicated Ydj1p-CaaaX variant were evaluated by SDS-PAGE and anti-Ydj1p immunoblot. The indicated Ydj1p variants were expressed in yWS2544 (ydj1::KANR) to eliminate any contribution from naturally encoded Ydj1p. Farnesylated Ydj1p (i.e., CASQ and CMIIM; closed triangle) exhibited a smaller apparent molecular mass relative to unmodified Ydj1p (i.e., SASQ and SMIIM; open triangle). Farnesylation profiles for the indicated Ydj1p-CaaaX variants were determined across multiple technical replicates, from which the percent of farnesylated species relative to the total signal for a sample was determined (see Appendix A and Appendix A).

**Table 1 ijms-22-12042-t001:** Summary of peptides observed in MALDI-MS libraries. Peptides chosen for further evaluation are bolded.

Library Sequence	Observed Amino Acid Hits
Ca_1_IIM	**S**, C, M, F, **Y**, A, P, Q, E, **H**
CMa_2_IM ^a^	G, S, N, **K**, Q, E, H, R
CMIa_3_M ^a^	**G**, N, M, A, T, L, Q, E, H
CMIIX	**S**, C, **K**, A, **Q**, M

^a^ The peptide CMIIM was also detected in this library, but with a signal-to-noise ratio of less than 10, and hence was not included in this tabulated data.

**Table 2 ijms-22-12042-t002:** Percent farnesylation of CaaaX peptides derived from MALDI libraries. Each value is the result of triplicate experiments.

	Extent of Conversion at	Extent of Conversion at
Sequence	25 nM yFTase, rt (%)	200 nM rFTase, 35 °C (%)
CMIIM	56 ± 10	50 ± 3
CMIIS	61 ± 14	25 ± 3
CMIIQ	76 ± 3	80 ± 2
CSIIM	64 ± 5	<1
CMKIM	54 ± 10	<1
CYIIM	95 ± 1	49 ± 1
CHIIM	15 ± 6	<1
CMIGM	<1	<1
CMIIK	<1	<1

**Table 3 ijms-22-12042-t003:** Percent farnesylation of CaaaX peptides derived from mammalian genome. Each value is the result of triplicate experiments. ND = not determined.

	Extent of Conversion at	Extent of Conversion at	Extent of Conversion at	Extent of Conversion at
Sequence	25 nM yFTase (%)	100 nM yFTase (%)	25 nM rFTase (%)	100 nM rFTase(%)
CSLMQ	95 ± 4	>99	79 ± 2	>99
CSQAS	43 ± 3	>99	ND	66 ± 1
CMTSQ	ND	62 ± 1	ND	9 ± 1
CASSQ	ND	33 ± 3	ND	<1
CQYNS	ND	<1	ND	<1
CLACS	ND	<1	ND	<1
CVQTS	ND	<1	ND	<1
CASLS	ND	<1	ND	<1
CSKLN	ND	<1	ND	<1
CLLFS	ND	<1	ND	<1

## Data Availability

Data is contained within the article and Appendix A, raw data are available on request from the corresponding author.

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
