# Peer review of "MALDI-MS Analysis of Peptide Libraries Expands the Scope of Substrates for Farnesyltransferase"

_ijms, 2021, doi:10.3390/ijms222112042_

Round 1

Reviewer 1 Report

  1. SPPS should be explained when first used
  2. In discussion explain why the matrix does not interfere at MW under 1,000
  3. Please explain why the ESI at higher Tesla would not accomplish the same

Reviewer 2 Report

In this manuscript by Schey et al., the authors develop complementary peptide libraries and establish MALDI-MS as tools to study protein prenylation. Using these novel tools and methods, the authors demonstrate convincingly that several novel variants of a 5 amino acid prenylation motif are indeed prenylated. The authors confirm these results by showing that the same motifs are modified in the context of Ydj1p, a yeast HSP40 chaperone. Overall, the study is well designed and the conclusions are supported by the results. Consequently, most of my points are minor and relate to semantics / language use.

Major comments:

  • Figure 5 shows convincingly how Ydj1p variants with distinct C-terminal CaaaX motifs are prenylated in vivo. The authors disclose a single blot in each panel and show a percentage of farnesylation. In the Materials section, the authors disclose that at least two technical replica were made of each blot. Ideally, this number should be at least 3. Further, a mean percentage value +/- standard deviation and appropriate statistical test should be disclosed in Fig.5. Finally, for the sake of transparency, images of all replica blots should be presented as supp. figure.
  •  

Minor comments:

  • Starting with the abstract, several abbreviations are not explained when used the first time. For example, SPPS is defined in the Materials and methods section, but appears in the abstract and throughout the manuscript. MALDI-MS & FPP isn't defined anywhere. Please define all abbreviations the first time they are used.
  • The authors use prenylation and farnesylation interchangeably, sometimes in the same sentence (e.g. lines 106-108). However, the introduction does not explain that both farnesylation and geranyl-geranylation are types of prenylation. Please edit the introduction section such that readers unfamiliar with prenylation as a whole also understand what farnesylation, geranyl-geranylation etc. refers to. For constancy reasons, I would further recommend to stick to either farnesylation or prenylation throughout the manuscript. 
  • lines 135-136: it is a bit surprising that increasing the enzyme concentration doesn't result in more efficient peptide modification. Comparing Figure 2 to Figure 1 clearly shows that it is not a saturation problem. This lack of a dose-response effect should at least be addressed in the discussion section.
  • Paragraph 2.5 should be edited such that it discloses that the authors expressed distinct Ydj1p versions in Yeast cells and then analyzed Yeast cell lysates by immunoblotting. 
  • lines 62-64: please reference the publications that established the presented kcat/Km values.

Language editing:

  • it appears that lines 28-30 use a different font type
  • line 55 (favored or unfavored should be preferred),
  • lines 68-70 don't read well
  • line 77: please replace envisioned with developed
  • line 83: please replace envisioned with hypothesized 
  • line 87-91 read as if this work wasn't done yet (planned on utilizing); please change
  • line 118 and others: Gratifyingly is unnecessary
  • line 119: add references (previous studies)
  • line 223: ability should be availability or accessibility
  •  
